# Longitudinal Assessment of Botulinum Toxin Treatment for Lateral Trunk Flexion and Pisa Syndrome in Parkinson’s Disease: Real-life, Long-Term Study

**DOI:** 10.3390/toxins15090566

**Published:** 2023-09-11

**Authors:** Claudia Ledda, Elisa Panero, Ugo Dimanico, Mattia Parisi, Marialuisa Gandolfi, Michele Tinazzi, Christian Geroin, Francesco Marchet, Giuseppe Massazza, Leonardo Lopiano, Carlo Alberto Artusi

**Affiliations:** 1Department of Neuroscience “Rita Levi Montalcini”, University of Turin, 10126 Turin, Italy; claudia.ledda@unito.it (C.L.); leonardo.lopiano@unito.it (L.L.); 2Neurology 2 Unit, AOU Città della Salute e Della Scienza, 10126 Turin, Italy; 3Physical Medicine and Rehabilitation Unit, Department of Orthopedics, Traumatology and Rehabilitation, University of Turin, 10126 Turin, Italy; elisa.panero@polito.it (E.P.); giuseppe.massazza@unito.it (G.M.); 4Department of Neurology, Ospedale Rivoli, Rivoli, 10098 Turin, Italy; mattia.parisi@unito.it; 5Neurology Unit, Movement Disorders Division, Department of Neurosciences Biomedicine and Movement Sciences, University of Verona, 37129 Verona, Italy; marialuisa.gandolfi@univr.it (M.G.); michele.tinazzi@univr.it (M.T.); christian.geroin@univr.it (C.G.); 6Neuromotor and Cognitive Rehabilitation Research Centre (CRRNC), University of Verona, 37129 Verona, Italy; 7Department of Human Neurosciences, Sapienza University of Rome, Viale dell’Università 30, 00185 Rome, Italy; francesco.marchet@uniroma1.it

**Keywords:** botulinum toxin, lateral trunk flexion, Pisa syndrome, Parkinson’s disease

## Abstract

Lateral trunk flexion (LTF) and its severe form, called Pisa syndrome (PS), are highly invalidating axial postural abnormalities associated with Parkinson’s disease (PD). Management strategies for LTF lack strong scientific evidence. We present a real-life, longitudinal study evaluating long-term efficacy of botulinum toxin (BoNT) injections in axial muscles to reduce LTF and PS in PD. A total of 13 PD patients with LTF > 5° received ultrasound- and electromyography-guided BoNT injections every 4 months. Seven untreated matched PD patients with LTF served as controls and their changes in posture after 18 months were compared with those of seven patients continuing BoNT over 12 months. 53.8% of patients continued the BoNT injections for at least 12 months. Various individual LTF responses were observed. Overall, BoNT-treated patients obtained a not statistically significant improvement of LTF of 17 ± 41% (*p* = 0.237). In comparison, the seven untreated PD patients suffered a deterioration in LTF over 12 months by 36 ± 45% (*p* = 0.116), showing a significantly different trajectory of posture change (*p* = 0.026). In conclusion, repeated BoNT injections in axial muscles showed varying effects in managing PD-associated LTF, suggesting that: (a) a relevant number of patients with LTF can benefit from BoNT; (b) long-term treatment could prevent LTF worsening; (c) an instrumented, personalized approach is important; and (d) there is a need for prospective, long-term studies.

## 1. Introduction

Patients with Parkinson’s disease (PD) often suffer from axial postural abnormalities, leading to greater motor impairment, severe back pain, increased risk of falling and disability, and reduced quality of life [1,2,3]. Among axial postural abnormalities, Pisa syndrome (PS), defined as a >10 degrees lateral trunk flexion appearing while sitting or standing, is an invalidating axial motor symptom observed in about 10% of PD patients [3,4]. Recently, a milder form of PS called ‘Lateral trunk flexion’ (LTF) has been defined by the International Parkinson and Movement Disorder Society (MDS) Task Force on Postural Abnormalities as an involuntary lateral bending of the trunk with an angle ranging from 5 to 10 degrees [5]. The pathophysiology of PS and LTF is likely multifactorial, and several central and peripheral mechanisms have been implicated, such as dystonia, impaired proprioception, basal ganglia dysfunction, abnormal sensorimotor integration, vestibular impairment, cognitive dysfunction, drugs, rigidity, muscle atrophy/fatty degeneration, and myopathy of paraspinal muscles [1,2,6,7,8,9,10,11,12].

The best management options of PS are not yet adequately supported by scientific evidence. Currently, the available options provided by expert opinion are: (a) revision of dopaminergic therapy, (b) rehabilitation programs, (c) botulinum toxin (BoNT) treatment, (d) deep brain stimulation (DBS), (e) spinal surgery [13,14]. Some literature findings suggest attempting treatment with BoNT injections in hyperactive paraspinal and non-paraspinal muscles. Indeed, few trials and case reports showed an improvement of PS after BoNT injections in paraspinal and non-paraspinal muscles [15,16,17,18,19]. However, due to the lack of standardization in the injection protocol and limited data regarding the muscles involved in axial postural abnormalities, the scientific evidence remains limited. Finally, to date, no longitudinal studies have been published on the long-term effect of BoNT in improving PS and LTF.

In this context, we present a real-life, longitudinal, observational study to analyze the short- and long-term efficacy of BoNT injections in axial muscles for treating both LTF (>5 and ≤10 degrees) and PS (>10 degrees) [5] in PD patients at our center. We aimed to analyze the rate of patients improving after the first, second, and third course of BoNT injections and the rate of patients opting for continuing the therapy over 1 year, comparing their degrees of lateral trunk flexion versus a cohort of PD untreated patients with PS, followed up in another center and serving as a control group.

## 2. Results

We included in the analysis 13 PD patients (eight males; seven females) with LTF (n = 5) and PS (n = 8) [5]. The mean age at baseline was 68.8 ± 9 years, with a PD duration of 12.7 ± 6.2 years. The baseline angle of flexion while standing in a relaxed trunk position was 11.2 ± 4.6 degrees, while the visual analogue scale (VAS) score for back pain was 6 ± 3 [20]. Table 1 shows the main clinical and demographic characteristics of the BoNT-treated patients prior to treatment (T0).

### 2.1. Long-Term Treatment Outcome

Out of the total 13 patients, 53.8% (n = 7) continued their BoNT treatment for over 12 months; 30.8% (n = 4) received a single treatment course, and 15.4% (n = 2) received two treatment courses. Among the six patients who discontinued treatment within the first two courses, two patients discontinued because they died from causes unrelated to PD, while the remaining four patients discontinued treatment due to lack of efficacy (i.e., no perceived improvement in posture and/or in back pain/discomfort).

When comparing the angle of flexion at T0 (prior to BoNT treatment) with the last follow-up (at least 1 year after starting BoNT treatment of paravertebral muscles), we observed a mild improvement (from 10.4 ± 2.8 degrees to 9.2 ± 6.4 degrees; *p* = 0.237), which is statistically not significant. Please refer to Figure 1 and Table 2 for a detailed representation of the angle of deviation at each available follow-up and to Figure 2 for a patient’s picture as an example.

Two patients exhibited a worsening lateral flexion angle at T1 (1 month after the first BoNT treatment), but by the last follow-up, their angles had improved. Four patients showed improvements both at T1 and the last follow-up. One patient experienced posture worsening both at T1 and at the last follow-up.

### 2.2. Untreated Control Group and Comparison with Long-Term BoNT-Treated Group

The untreated control group consisted of seven PD patients with PS (four males; three females) matched with the seven PD patients treated with BoNT for over 12 months. Their mean age at baseline was 69.7 ± 3.2 years, with a PD duration of 7.7 ± 5.8 years. The baseline angle of trunk flexion while standing in a relaxed trunk position was 15.1 ± 4.3 degrees. Table 3 shows the main clinical and demographic characteristics of PS untreated controls and long-term treated patients.

The two groups did not significantly differ in motor impairment (measured using the Movement Disorder Society Unified Parkinson’s Disease Rating Scale—MDS UPDRS—part III) [21], years of PD and lateral trunk flexion duration, age, and other clinical/demographical characteristics. However, there was a significant difference in baseline flexion degrees, which were higher in the control group (*p* = 0.038) (Table 3).

The angles of lateral trunk flexion in the control group changed from 15.1 ± 4.3 degrees at baseline to 20.2 ± 7.4 degrees in the follow-up (*p* = 0.116) (Figure 3). The comparison of changes of lateral trunk flexion degrees over time between the two groups (treated and untreated control group) was significantly different (*p* = 0.026), with degrees of untreated control group deteriorating by an average of 36%, and those of BoNT-treated group improving of an average of 17%.

### 2.3. Short-Term Treatment Outcome

Comparing variables at T0 and T1, at a group level, we did not observe a significant change in the angle of lateral trunk flexion in a relaxed trunk position (from 11.2 ± 4.6 degrees to 12.4 ± 7.5 degrees; *p* = 0.507). The VAS score changed from 6 ± 3 to 5 ± 3.3 (*p* = 0.606), and the quality of life measured using the Eight-Item Parkinson’s Disease Questionnaire (PDQ-8) [22] from 23.6 ± 20.3 to 19.8 ± 14 (*p* = 0.878).

The clinical and demographical characteristics at baseline of patients who continued BoNT treatment long-term and those who underwent only one or two treatment cycles are available in Table 4. The duration of LTF or PS was different between long- and short-term groups (2.4 ± 2.2 vs. 6.2 ± 6.8; *p* = 0.057), albeit not significant, while the T0 lateral trunk flexion angles were similar between groups (10.4 ± 2.8 degrees vs. 12.2 ± 6.2 degrees; *p* = 0.886).

## 3. Discussion

This is the first study reporting the long-term use of BoNT for treating PS and LTF in PD. Our findings, based on real-world observational data, indicate that about half of patients treated with BoNT opted in accordance with the clinician to continue to be treated for over 12 months. Moreover, this study results suggest that patients who benefit the most and choose to continue BoNT treatment are those with a shorter duration of trunk flexion. Finally, we found that patients treated over 12 months did not significantly worsen their posture over time, while a similar untreated control group of PD patients with PS showed a worsening of their posture over time.

Few prior studies have reported promising outcomes using BoNT for PS patients. Bonanni et al. conducted a blind cross-over study with BoNT and placebo in nine PD patients with lateral axial dystonia, defined as a lateral trunk flexion >15 degrees toward one side, which increased during walking and disappeared in the recumbent position, characterized by continuous electromyographic (EMG) activity of lumbar paraspinal muscles ipsilateral to the bending side [15]. 125 UI of Abobotulinumtoxin-A was injected under EMG-guidance into four sites in the paraspinal muscles, 2 to 2.5 cm lateral to spinous processes at level L2-L5 on the side of the trunk flexion for a total dose of 500 U. No patient reported benefit after the placebo, while treatment was effective in six patients, and four out of six patients continued to receive BoNT for 2 years after the study [15].

In a randomized placebo-controlled trial, Tassorelli et al. enrolled 26 PD patients with PS (>10 degrees of lateral trunk flexion) [16]. The experimental group (group A) was treated with incoBoNT-A before the 4-week rehabilitation program. The control group (group B) received saline before rehabilitation. The injection was performed if the EMG pattern was characterized by involuntary tonic activity longer than 500 ms. The Authors treated < 6 sites per patient, with a maximum dose of 50 UI per site and a total dose per patient of between 50 and 200 UI. They demonstrated that PS was significantly reduced versus baseline in patients receiving BoNT treatment compared to placebo [16].

Another study evaluated the efficacy of BoNT-A under MRI, US- and EMG-guided injections in 15 PS patients [19]. PS was defined as a lateral trunk flexion of at least 10 degrees improved by passive mobilization and supine positioning, and the injection protocol consisted of OnaBoNT-A (dissolved as 100 units in 2 mL of saline) injected under US and EMG guidance in muscles with pathological muscular hyperactivity on EMG study when standing and absence of severe muscle atrophy on MRI. Out of the 13 patients available for follow-up, they have reported that 84.6% of patients (11 out of 13) ameliorated at least 5 degrees, while pain improved in all patients [19]. In addition, two other case reports showed an improvement of posture in PS after BoNT treatment [17,18].

Altogether, these studies show the potential of BoNT injection in treating PS and LTF in PD. However, these studies were based on the analysis of posture after one course of BoNT treatment; moreover, some variability in outcomes has been observed, possibly due to the difference in treatment approaches. Indeed, the three studies differed for treatment dosages, dilution levels, injection sites, and use of ultrasound guidance. Furthermore, the differing definitions of lateral trunk flexion have led to the inclusion of heterogeneous patient samples, partially contributing to the difference in results. The MDS Task Force on Postural Abnormalities has recently defined cut-off values to standardize the definition of axial postural abnormalities on the coronal plane, dividing the ‘Lateral trunk flexion’ (≥5 degrees to ≤10 degrees) from its severe form, which is called ‘PS’ (>10 degrees) [5]. Shared terminology and cut-offs are crucial for early identification and estimation of the axial postural abnormality to prompt early interventions [5,13]. Our findings further support this notion by suggesting that individuals with a shorter duration of LTF or its severe form (i.e., PS) may obtain the greatest benefits from BoNT treatment [23]. Data obtained from our observational study endorse the current expert opinion hypothesis on the relevance of early intervention before the axial postural abnormality reaches the threshold of degrees to be diagnosed with PS [13]. Noteworthy, the group of patients treated with BoNT in the long term did not experience deterioration of posture over time, while a longitudinal untreated control group of patients with PS with similar clinical and demographic characteristics showed a significant worsening of the lateral trunk flexion over time. The small sample size and the retrospective study design notwithstanding, these data suggest that BoNT treatment offered in LTF might act as a measure to prevent further worsening of lateral trunk flexion, even in case of lack of improvement.

The muscles to target and the dosages of BoNT in this study have been chosen based on previous literature and our center’s experience in treating axial hyperactive muscles (e.g., dystonic patients and PD patients with postural abnormalities). However, other muscles, such as abdominal muscles and quadratus lumborum, have been previously indicated as involved in the postural abnormality related to PD-associated PS [24]. Another controversial aspect can be related to the exact point for muscle injection, considering that both longissimus thoracis and iliocostalis lumborum are long muscles [25,26]. We opted to inject at the level of maximal spine flexion (i.e., fulcrum), but the possibility of performing multiple injections in the same muscles could be considered.

As anticipated, our study has several limitations, including the small sample size and a retrospective study design. However, trunk flexion degrees were assessed by objective measurement, and the fact that it is a real-life study with longitudinal, long-term data on both treated and untreated PD patients with LTF or PS is a novel contribution to the existing literature. It is important to consider in the interpretation of results three other main aspects: (1) except for the follow-up assessment performed 1 month after the first BoNT injection, all other evaluations of BoNT-treated patients were performed 4 months after the BoNT injection, possibly reducing our ability to detect short-term improvement provided by the treatment; (2) the control group is based on patients with a diagnosis of PS, thus showing higher degrees of trunk flexion at baseline than the BoNT treated group: whether this aspect impacted the progression of trunk flexion over time is difficult to know; (3) the criteria for defining a clinically significant improvement in degrees of lateral trunk flexion still need to be established. Previous studies on PS considered 5° as a potentially significant change [16,19,27]; however, this is an arbitrary cut-off not supported by clinical evidence and it seems not suitable in patients with LTF, for which smaller changes could be of clinical importance.

## 4. Conclusions

In conclusion, a longitudinal randomized controlled trial with adequate sample size and standardized definitions of LTF and injection procedures will be necessary to obtain strong evidence on the efficacy of BoNT in patients with LTF and PS. Meanwhile, we believe our data contribute to supporting the use of BoNT injections for treating PD-related LTF and PS. Considering our findings and the recent position paper of experts in the field [13], we recommend to: (a) pay attention to early cases and indicate early treatment before the development of severe trunk flexion, and (b) prefer a personalized approach based on EMG and US guidance for appropriate muscle targeting rather than a standardized injection protocol.

## 5. Materials and Methods

### 5.1. Study Population

Consecutive outpatients with a diagnosis of idiopathic PD complicated by LTF (defined as a reversible flexion of the trunk in the coronal plane between 5 and 10 degrees) [5], or PS (defined as a reversible flexion of the trunk in the coronal plane > 10 degrees) [5], were evaluated for treatment with BoNT injection by a movement disorder expert of Città della Salute e della Scienza di Torino, Molinette Hospital, University of Turin, between January 2021 and January 2022. Inclusion criteria were: a diagnosis of idiopathic PD according to the Movement Disorders Society criteria [28], the presence of involuntary lateral trunk flexion of >5 degrees which could be improved by passive mobilization and supine positioning [5], a Hoehn and Yahr (H&Y) stage [28] between II and IV, and the ability to walk unaided for at least one minute. Exclusion criteria included a history of spine surgery or orthopedic diseases known to affect posture, and a diagnosis of dementia, defined as a Mini-Mental State Examination score of <24/30 [29].

### 5.2. Clinical Assessment

As per the clinical practice routine in our center, patients underwent total spine radiography in a standing and a dorsal-lumbar spine Magnetic resonance Imaging (MRI) (1.5 Tesla), with sequences allowing the assessment of the paravertebral, abdominal, and iliopsoas muscles.

Patients’ motor disability was evaluated by means of the Movement Disorder Society Unified Parkinson’s Disease Rating Scale (MDS-UPDRS) part III, while non-motor and motor experience of daily living impairment was assessed by MDS-UPDRS part I and II, respectively, and motor fluctuations by MDS-UPDRS part IV [21]. Health-related quality of life was assessed with the PDQ-8 and back pain was measured using the VAS [20,22]. Demographic data and all medications were recorded, and the levodopa equivalent daily dose (LEDD) was calculated using a validated conversion formula [30].

### 5.3. Posture Assessment

Standardized pictures were captured for each patient in coronal view while standing in a relaxed trunk position. LTF or PS degrees were determined from pictures using the validated NeuroPostureApp according to the perpendicular method (the angle between (a) the line connecting the midpoint feet with L5 or the pubic symphysis and (b) the line connecting the pubic symphysis with the jugulum or L5 and C7 spinous processes) [5,31,32]. We repeated this iconographic evaluation for all patients one month after the initial BoNT injection and before each subsequent treatment course.

The ability to revert the posture misalignment by passive maneuver or in a supine lying position was verified in all patients before the first BoNT treatment.

### 5.4. Ultrasound- and Electromyography-Guided BoNT Injection

Before the first BoNT course, electrophysiological activity of the following muscles on both sides were assessed by needle electromyography (Medtronic Keypoint portable 4-channel electromyography) under the guidance of ultrasonography (US, Alpinion E-Cube 8) [33]: longissimus, iliocostalis (at lumbar and thoracic levels adjusted for each patient according to the fulcrum of spine curvatures), internal and external abdominal oblique muscles. Muscles were evaluated with patients under different testing conditions: (i) while sitting without any upper limb or trunk support (with the patient’s trunk maintained in a relaxed position); (ii) during a trunk passive mobilization (recording the muscle activity with the trunk maintained in an upright straight position by an operator); and (iii) during self-realignment of the trunk, with a specific maneuver normally activating the muscle under evaluation. When muscles showed EMG activity at rest, maneuvers of muscle deactivation were used and recorded (antagonists’ activation). The motor unit action potentials (MUAPs) were visually inspected for each testing condition.

The patient was then treated at the beginning (T0) by injection of 100 Units of Onabotulinum toxin-A (OnaBonT-A/Botox^®^), diluted in 2 mL of saline under US and EMG guidance, in the longissimus thoracis and the iliocostalis lumborum from the same side of the LTF (50 UI for each muscle). All patients were reassessed with the same methodology 1 month after BoNT injection (T1). The patients were then reassessed 4 months (T2), 8 months (T3), and 12 months (T4) after the first BoNT injection (T0), with the same methodology, before the following courses of BoNT treatment.

Starting from T2, all treatments were personalized for each patient under the guidance of US and EMG if satisfactory results were not achieved after the initial standardized injection course. Specifically, BoNT injections were then performed in longissimus thoracis and/or iliocostalis lumborum muscles with pathological muscular hyperactivity on EMG when sitting without support. When muscular hyperactivity was observed bilaterally, bilateral injections were also performed, typically with higher doses reserved for the more active muscle. Doses of OnaBonT-A varied from 15 to 50 units for each injected muscle (Figure 4).

## Figures and Tables

**Figure 1 toxins-15-00566-f001:**
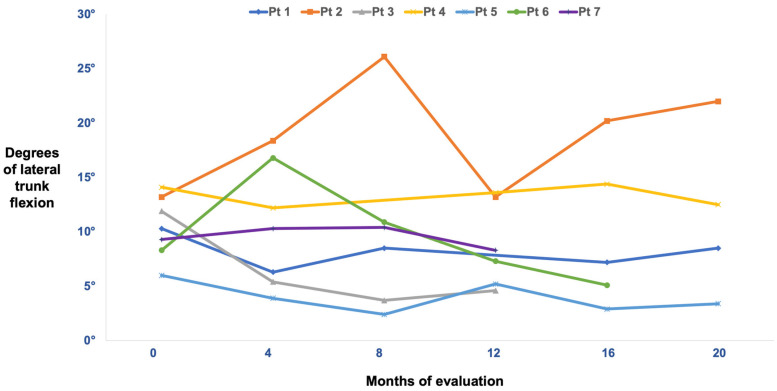
Long-term individual assessment of LTF degrees in BoNT-treated patients. Patient (Pt).

**Figure 2 toxins-15-00566-f002:**
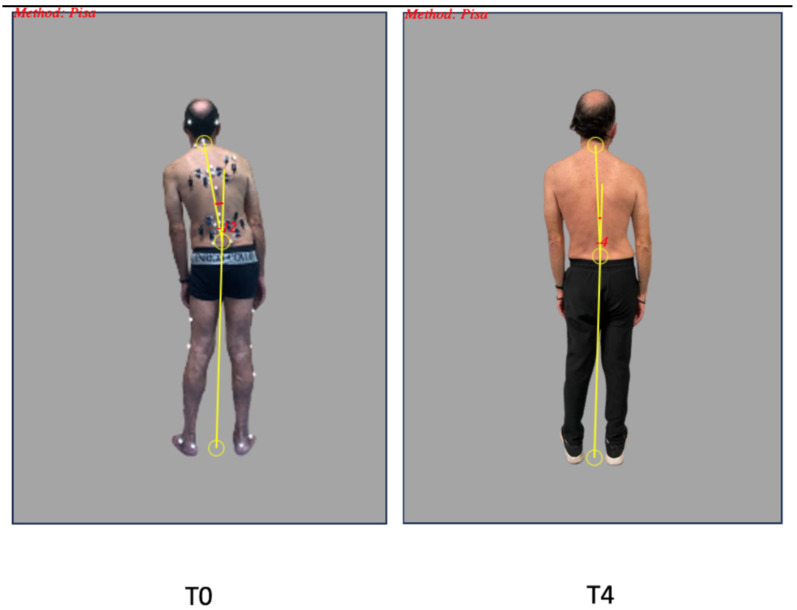
Assessment of LTF degrees in a long-term BoNT-treated patient. The degree calculation was performed with the NeuroPostureApp (http://www.neuroimaging.uni-kiel.de/NeuroPostureApp/, accessed on 1 May 2023).

**Figure 3 toxins-15-00566-f003:**
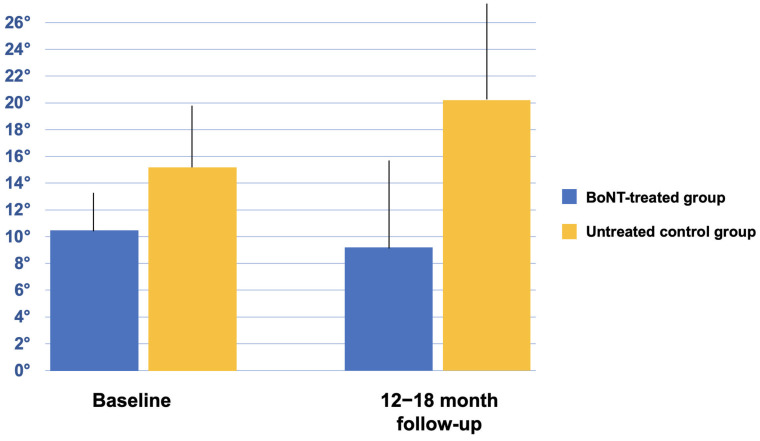
Degrees of trunk flexion of BoNT-treated group vs. untreated control group.

**Figure 4 toxins-15-00566-f004:**
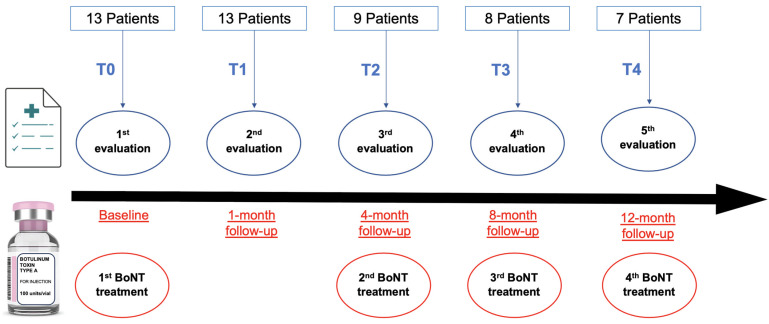
Treatment and follow-up of the study sample.

**Table 1 toxins-15-00566-t001:** Clinical and demographical characteristics of the BoNT-treated patients at baseline (T0).

	BoNT-Treated Patients (n = 13) Mean ± SD
Age (yrs)	68.8 ± 9
PD duration (yrs)	12.7 ± 6.2
Age at PD onset (yrs)	56.1 ± 11.9
Age at PS onset (yrs)	64.8 ± 11.3
PS duration (yrs)	4.1 ± 5
Latency PD onset, PS onset (yrs)	9.4 ± 6
MDS-UPDRS III	39.8 ± 15.3
MDS-UDPRS IV	2.5 ± 2.5
Angle of LTF at baseline (degree)	11.2 ± 4.6
VAS score	6 ± 3
PDQ-8 score	23.6 ± 20.3
Total LEDD (mg)	933.9 ± 390.5
Levodopa LEDD (mg)	723.1 ± 359.6
Dopamine Agonist LEDD (mg)	158.8 ± 116.3

Botulinum Toxin (BoNT); Parkinson’s Disease (PD); Pisa syndrome (PS); Movement Disorder Society Unified Parkinson’s Disease Rating Scale (MDS-UPDRS); Lateral Trunk Flexion (LTF); Visual Analogue Scale (VAS); The Parkinson’s Disease Questionnaire (PDQ-8); Levodopa equivalent daily dose (LEDD).

**Table 2 toxins-15-00566-t002:** Treatment and outcome of long-term follow-up patients at different time points.

Pt		Time 0	4 Months	8 Months	12 Months	16 Months	20 Months
1	Angle	10.3°	6.3°	8.5°	NA	7.2°	8.5°
	Side of trunk flexion	Right	Right	Right	Right	Right	Right
	Treatment	Right IL 50U Right LT 50U	Right IL 50U Right LT 50U	Right IL 50U Right LT 50U	Right IL 50U Right LT 50U	Right IL 50U Right LT 50U	Right IL 50U Right LT 35U Left LT 15U
2	Angle	13.2°	18.4°	26.1°	13.2°	20.2°	22°
	Side of trunk flexion	Right	Right	Right	Right	Right	Right
	Treatment	Right IL 50U Right LT 50U	Left IL 50U Left LT 50U	Left IL 50U Left LT 50U	Left IL 50U Left LT 50U	Left IL 50U Left LT 50U	Left IL 50U Left LT 50U
3	Angle	11.9°	5.4°	3.7°	4.6°	NA	NA
	Side of trunk flexion	Left	Left	Left	Left	Left	Left
	Treatment	Left IL 50U Left LT 50U	Left IL 50U Left LT 50U	Left IL 50U Left LT 50U	Left IL 50U Left LT 50U	Left IL 40U Left LT 40U Right IL 15U Right LT 15U	Left IL 40U Left LT 40U Right IL 15U Right LT 15U
4	Angle	14.1°	12.2°	NA	13.6°	14.4°	12.5°
	Side of trunk flexion	Right	Right	Right	Right	Right	Right
	Treatment	Right IL 50U Right LT 50U	Right IL 50U Right LT 50U	Right IL 50U Right LT 50U Left IL 20U Left LT 20U	Right IL 50U Right LT 50U Left IL 20U Left LT 20U	Right IL 40U Right LT 30U Left IL 15U Left LT 15U	Right IL 40U Right LT 30U Left IL 15U Left LT 15U
5	Angle	8.3°	16.8°	10.9°	7.3°	5.1°	NA
	Side of trunk flexion	Right	Right	Right	Right	Right	Right
	Treatment	Right IL 50U Right LT 50U	Right IL 15U Right LT 35U Left IL 50U Left LT 35U	Right IL 15U Right LT 35U Left IL 50U Left LT 35U	Right IL 15U Right LT 35U Left IL 50U Left LT 35U	Right IL 15U Right LT 35U Left IL 50U Left LT 35U	Right IL 15U Right LT 35U Left IL 50U Left LT 35U
6	Angle	6°	3.9°	2.4°	5.2°	2.9°	3.4°
	Side of trunk flexion	Left	Left	Left	Left	Left	Left
	Treatment	Left IL 50U Left LT 50U	Left IL 50U Left LT 50U	Left IL 50U Left LT 50U	Left IL 50U Left LT 50U	Left IL 50U Left LT 50U	Left IL 50U Left LT 50U
7	Angle	9.3°	10.3°	10.4°	8.3°	NA	-
	Side of trunk flexion	Right	Right	Right	Right	Right	
	Treatment	Right IL 50U Right LT 50U	Right IL 50U Right LT 50U Left IL 20U Left LT 20U	Right IL 50U Right LT 50U Left IL 20U Left LT 25U	Right IL 50U Left IL 20U Left LT 25U	Right IL 50U Left IL 25U Left LT 25U	

Patient (Pt); Iliocostalis lumborum (IL); Longissimus thoracis (LT); Patients treated but angles of LTF not available (NA); Patients not treated (-).

**Table 3 toxins-15-00566-t003:** Clinical and demographical characteristics of the long-term BoNT-treated patients and untreated control patients at baseline (T0).

	Long-Term BoNT-Treated Patients (n = 7) Mean ± SD	Untreated Control Patients (n = 7) Mean ± SD	*p* Value
Age (yrs)	67 ± 10.7	69.7 ± 3.2	0.805
PD duration (yrs)	12.3 ± 6.6	7.7 ± 5.8	0.209
Age at PD onset (yrs)	54.7 ± 13.3	59.8 ± 6.9	0.805
Age at PS onset (yrs)	64.8 ± 10.8	66 ± 4.4	1
PS duration (yrs)	2.4 ± 2.2	3.7 ± 3.3	0.620
Latency PD onset, PS onset (yrs)	10.8 ± 6.4	6.2 ± 4.3	0.097
MDS-UPDRS III	34.7 ± 12.2	31.6 ± 8.2	0.805
MDS-UDPRS IV	2.4 ± 2.2	1.3 ± 2.2	0.318
Angle of LTF at baseline (degree)	10.4 ± 2.8	15.1 ± 4.3	0.038 *
VAS score	6 ± 3.3	5.8 ± 2.6	0.818
PDQ-8 score	19.2 ± 13.3	28.6 ± 13.8	0.209
Total LEDD	804.2 ± 249.7	860.4 ± 302.6	0.805

Botulinum Toxin (BoNT); Parkinson’s Disease (PD); Pisa syndrome (PS); Movement Disorder Society Unified Parkinson’s Disease Rating Scale (MDS-UPDRS); Lateral Trunk Flexion (LTF); Visual Analogue Scale (VAS); The Parkinson’s Disease Questionnaire (PDQ-8); Levodopa equivalent daily dose (LEDD); statistically significant difference (*).

**Table 4 toxins-15-00566-t004:** Clinical and demographical characteristics of the long-term follow-up vs. short-term follow-up patients.

	Long-Term Follow-Up Patients (n = 7/13) Mean ± SD	Short-Term Follow-Up Patients (n = 6/13) Mean ± SD	*p* Value
Age (yrs)	67 ± 10.7	71 ± 7	0.886
Age at PD onset (yrs)	54.7 ± 13.3	57.8 ± 11	0.775
Age at PS onset (yrs)	64.9 ± 10.8	64.8 ± 12.9	0.943
PS duration (yrs)	2.4 ± 2.2	6.2 ± 6.8	0.057
Latency PD onset, PS onset (yrs)	10.8 ± 6.4	7.7 ± 5.8	0.473
MDS-UPDRS III	34.7 ± 12.2	45.7 ± 17.6	0.253
MDS-UDPRS IV	2.4 ± 2.2	2.6 ± 3.1	1
Angle of trunk flexion at baseline—Relax (degree)	10.4 ± 2.8	12.2 ± 6.2	0.886
Baseline VAS score	6 ± 3.3	6.2 ± 2.9	0.935
Total LEDD	804.2 ± 249.7	1085.2 ± 490.1	0.252

Botulinum Toxin (BoNT); Parkinson’s Disease (PD); Pisa syndrome (PS); Movement Disorder Society Unified Parkinson’s Disease Rating Scale (MDS-UPDRS); Visual Analogue Scale (VAS); The Parkinson’s Disease Questionnaire (PDQ-8); Levodopa equivalent daily dose (LEDD).

## Data Availability

The data supporting this study’s findings are available from the corresponding author upon reasonable request.

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
