# Peer review of "Longitudinal Assessment of Botulinum Toxin Treatment for Lateral Trunk Flexion and Pisa Syndrome in Parkinson’s Disease: Real-life, Long-Term Study"

_toxins, 2023, doi:10.3390/toxins15090566_

Round 1
Reviewer 1 Report
1. Figures and tables look too drab, more color should be added to make it more beautiful and comfortable to read. Also, the font size of Figure 1 can be increased to facilitate reading. The typesetting format of Table 2 needs to be improved. Some characters are not in one line, which looks very ugly.
2. In the disccusion part, it's better to analyze the reason of the inconsistent outcomes of the different studies. Also, your own result should be more highlighted, and the significance and outlook of this study should be mentioned.
3. The results should be presented more clearly. Considering that the sample size is limited, maybe some additional measurements and experiments should be done to validate your conclusion.
There are some minor problems in the grammar and tense of the article. It is suggested that the author check and revise it carefully to make it conform to the norms of academic papers.
Reviewer 2 Report
Dear authors.
The study focuses on lateral trunk flexion and its severe form, Pisa syndrome, both significant postural abnormalities in Parkinson's disease. Currently, management strategies lack strong scientific backing. The study presents a real-life, longitudinal assessment of the effectiveness of botulinum toxin injections in axial muscles
I would recommend having a patient before after the treatment photos.
And It seems like the injection were based on needle electromyography) and by the guide of ultrasonography. Were electromyography helped you with the injection points? Because lots of cases in spasticity and involuntary contraction it is hard to point out injection point especially these long muscles. “Limitations of Electromyography in the Assessment of Abdominal Wall Muscle Contractility Following Botulinum Toxin A Injection”
Why did you choose these muscles for targeting? longissimus, iliocostalis. There is publications on injection points regarding neuromuscular junctions. Refer to these articles in the discussion. “Guidance to trigger point injection for treating myofascial pain syndrome: Intramuscular neural distribution of the quadratus lumborum” and “Effective botulinum neurotoxin injection in treating iliopsoas spasticity. In case of skeletal muscles, there needs to have delicate injection since larger doses are injected causes financial problems for patients. And why did you use these units for targeting them.
Nothing to comment about it.
Reviewer 3 Report
The number of patients is low and statistical analysis is not possible, put as a real world data paper the subject is of interest.I would change the title, as the treatment is not a standardized one and the results are coming just from an observational manner.
Round 2
Reviewer 1 Report
well revised, no more comments.
Reviewer 2 Report
Everything has been revised accordingly.
nothing to declare